# Antitumor Activities of Interleukin-12 in Melanoma

**DOI:** 10.3390/cancers14225592

**Published:** 2022-11-14

**Authors:** Wei Gao, Jun Pan, Jianping Pan

**Affiliations:** 1Institute of Translational Medicine, Zhejiang University City College, Hangzhou 310015, China; 2Institute of Cancer, Second Affiliated Hospital, Zhejiang University School of Medicine, Hangzhou 310009, China

**Keywords:** melanoma, IL-12, cancer therapy

## Abstract

**Simple Summary:**

Immunotherapy has been evolving rapidly in recent years. As a key component of the immune system, cytokines play essential roles not only in immune regulation but also in anti-tumor immunity. In the present review, we summarized the basic biological activities of IL-12 and its application in clinical trials for the therapy of melanoma. We also discussed the combined therapy of IL-12 with immune checkpoint inhibitors, cytokines and other therapeutic drugs in preclinical and clinical studies. The immune-related side effects of IL-12 and promising technological approaches that improve the efficacy and safety of IL-12 or IL-12 encoding vectors in the treatment of melanoma are also discussed.

**Abstract:**

Melanoma is the most common and serious malignant tumor among skin cancers. Although more and more studies have revolutionized the systematic treatment of advanced melanoma in recent years, access to innovative drugs for melanoma is still greatly restricted in many countries. IL-12 produced mainly by antigen-presenting cells regulates the immune response and affects the differentiation of T cells in the process of antigen presentation. However, the dose-limited toxicity of IL-12 limits its clinical application. The present review summarizes the basic biological functions and toxicity of IL-12 in the treatment of melanoma and discusses the clinical application of IL-12, especially the combination of IL-12 with immune checkpoint inhibitors, cytokines and other therapeutic drugs. We also summarize several promising technological approaches such as carriers that have been developed to improve the pharmacokinetics, efficacy and safety of IL-12 or IL-12 encoding plasmid application.

## 1. Introduction

As the most serious form of skin cancer and the most commonly diagnosed malignancy, the annual incidence of melanoma has been dramatically increasing during the past 50 years [1]. Prior to 2011, chemotherapy was the core of medical management for melanoma. In the past 10 years, there have been a variety of systematic therapies for melanoma that use a multitude of approaches, including molecular targeted therapy. Before the era of immunotherapy, patients with melanoma were treated using chemotherapy based mainly on BRAF and MEK inhibitors such as Vemurafenib and Binimetinib [2], or dacarbazine, temozolomide or fotemustine [3]. However, it is difficult to achieve the expected results, such as eliminating patients’ discomfort or prolonging survival time [4]. Given the circumstances, immune-checkpoint inhibitors (ICI) impressively showed more-durable responses, which have drastically improved the outcomes of patients with melanoma. The use of these ICIs, such as antibodies targeting PD-1 and its ligand, PD-L1, has expanded exponentially [5]. However, despite the effectiveness of ICIs, the number of patients who are able to achieve durable responses is restricted due to resistance to these immune agents; immune-related adverse events (irAE) complicate treatment [6].

Cytokines play an important role in the interactions between immune and non-immune cells within the tumor microenvironment. As a member of cytokines, interleukins are closely associated with the progression of cancer [7,8,9]. With the deepening of research on the mechanisms of interleukins’ regulation on tumor immune control and escape, the development of novel and more effective therapeutic strategies are promoted. The therapeutic potential of interleukins has been explored in both translational and basic cancer studies in recent years [10]. Researchers have been using molecular biology research and clinical trials to explore new and more effective methods to detect and treat melanoma more accurately. During this process, the therapeutic potential of interleukin as a single drug or as a combination of other types of clinical therapy for melanoma was highlighted. In this review, we summarized the basic mechanisms of IL-12 as well as its clinical applications in melanoma therapy. We also provide an up-to-date overview of clinical trials of IL-12 in melanoma patients over the past decade.

## 2. Major Molecular Regulation of IL-12 Signaling

IL-12 produced by monocytes serves as an effective inducer for T lymphocytes and NK cells to produce interferon-γ [11]. The IL-12 molecule itself is composed of two subunits: p35 and p40. The bioactive IL-12p70 is formed by the disulfide bond of these two subunits, and the p35 and p40 need to be co-expressed in the same cell to secrete bioactive IL-12p70 [12]. IL-12 can be secreted by different types of hematopoietic cells; however, it is mainly secreted by antigen-presenting cells, such as macrophages and dendritic cells (DC). IL-12 exerts its biological function via binding to its high-affinity receptor (IL-12Rβ1/IL-12Rβ2) hereby activating downstream tyrosine kinase 2 (TYK2), JAK2 and signal transducer and transcriptional activator (STAT) signaling pathways in T cells, NK cells and DCs. In DCs, blockade of just the IL-12p35 subunit effectively suppresses the tyrosine phosphorylation of JAK2, TYK2, STAT3 and STAT4 proteins, leading to the inhibition of cognate T cell response [13].

IL-12 secretion by dendritic cells supports the differentiation of CD4^+^ T cells into Th1 cells via augmenting the expression of T-bet [14]. IL-12 also promotes the proliferation of NK cells and NKT cells as well as their cytotoxicity, expression of cytotoxic mediators, and the production of cytokines, especially IFN-γ. Directly or with the help of IFN-γ, IL-12 activates the immune function of B cells with more production of IgG2a [15].

Ectopic T-bet expression induced by STAT1 activation selectively induced the expression of IL-12RB2 [16]. Interestingly, a designed panel of IL-12 partial agonists targeting the IL-23R/IL-12RB1 complex preserved IFN-γ production from CD8^+^ T cells, while impairing cytokine production from NK cells, due to IL-12RB1 being shared by IL-23 and IL-12 as a receptor signaling subunit [17]. The IL-12 signaling pathway and its major regulatory bioactivity are shown in Figure 1.

## 3. Anti-Tumor Effects of IL-12 in Melanoma

Immunotherapy is a promising modality to treat cancer. More and more studies have shown that IL-12 is a potent antitumor therapeutic agent for the treatment of many types of cancers in human patients, murine and even canine and horses [18]. Besides melanoma, IL-12 has been evaluated as an experimental treatment for numerous malignancies, such as lung cancer [19], colon cancer [20] and acute myelocytic leukemia (AML) [21].

Though the precise mechanisms underlying IL-12 antitumor effects need to be further clarified, growing evidence demonstrated that IL-12 mediated its antitumor effects via the regulation of many types of immune cells. IL-12 from CD103^+^ DCs is crucial for NK cell-mediated control of metastatic melanoma [22]. IL-12 inhibits melanoma progression via increasing the infiltration of NK cells and CD8α^+^ T cells, and a decreased presence of CD4^+^Foxp3^+^ regulatory T cells [23]. It preserves the function of CD8^+^ T cells via downregulating the expression of PD-1 as well as impairing the upregulation of PD-L1 expression induced by IFN-γ in tumor stromal cells, which is considered as negative regulation by local IFN-γ [24]. The frequency of inhibitory CD4^+^CD25^bright+^CD27^+^ Treg cells in metastatic melanoma patients was significantly decreased with IL-12 treatment [25]. The adoptive transfer of tumor-specific T cells has demonstrated satisfying clinical response [26]. IL-12 primed adoptive T cells were shown to be resistant to PD-1/PD-L1 mediated suppression and they sustained contact with intratumoral DCs [27]. Meanwhile, it was observed that the expression of PD-1 and LAG3 was suppressed after IL-12 treatment in human tumor-specific T cells [28]. The antitumor mechanisms of IL-12 are shown in Figure 2.

However, in most of the patients who suffer from melanoma, immunotherapies conducted as a single agent are not capable of triggering complete responses. What is more dissatisfying is that even partial responses in most patients with extensive disease or in patients who were pretreated heavily can hardly be observed [29]. As a cytokine for cancer immunotherapy, IL-12 provides a critical bridge between adaptive and innate immunity [30]. Studies of IL-12 in combination with tumor-specific therapeutic strategies showed an acceptable toxicity profile with promising evidence of clinical benefits in metastatic melanoma patients.

### 3.1. Clinical Administration of IL-12 in Patients with Melanoma

Based on the evidence from in vivo and in vitro studies, clinical trials were conducted to evaluate IL-12 therapeutic schedules (Table 1). A phase I trial study (NCT00683670) has reported the cases of vaccination with CD40L/IFN-γ-matured, IL-12p70-producing DCs against stage IV melanoma. In this trial, six of seven treated patients developed durable T cell immunity to all three melanoma gp100 antigen-derived peptides. For melanoma patients, DC vaccine-derived IL-12p70 levels positively correlated with time to progression, as did Tc1 immunity [31]. Other than the vaccine treatment, autologous tumor-infiltrating lymphocytes (TIL) were also evaluated after being transduced with a gene encoding a single chain IL-12 in metastatic melanoma in a phase I trial study (NCI 1-C-011) [32]. Intratumoral gene electroporation uses electric charges to help plasmid DNA enter into target cells reproductively and effectively, especially in metastatic melanomas [33]. The use of tavokinogene telseplasmid encoding IL-12 (Tavo) was evaluated in three phase II trials [34,35,36]. Algazi and colleagues demonstrated the good tolerability of intratumoral Tavo in melanoma; however, increased adaptive immune resistance was also observed in the trial [34]. In Greaney and colleagues’ study [35], the same treatment was performed in melanoma patients as in Algazi’s study [34]; the frequency of circulating PD-1 expressing CD4^+^ and CD8^+^ T cells was reduced, which indicated the high effectiveness of intratumor IL-12 electroporation on impacting circulating T cell via immunosuppression within the tumor microenvironment. Due to the low effectiveness of monotherapy with IL-12 or engineered IL-12 variants in melanoma patients or animal models, it is necessary to conduct more preclinical and clinical studies to evaluate the efficacy and safety of combination therapies with IL-12 or its variants. A phase II clinical trial of IL-12 therapy in combination with pembrolizumab (PD-1 inhibitor) (NCT02493361) was conducted and showed improved outcomes [36], which is described in the following section of IL-12 in combination with ICI.

### 3.2. IL-12 Therapy in Combination with ICI

ICI therapy has revolutionized the treatment of patients with melanoma; however, most of the patients are not able to achieve effective responses after checkpoint inhibition (CPI) therapy [37]. Although IL-12 yielded dramatic augmentation in immune cell infiltration, it has to be administrated in combination with other anti-cancer agents because of the inhibitory effect of immunological checkpoints. B16F10 melanoma responds poorly to CPI therapy. Treating the animals with CPI (anti-PD-1 and anti-CTLA-4 antibodies) alone showed just little effect on the growth of B16F10 tumors; meanwhile, the collagen-binding domain fused to IL-12 (CBD-IL-12) alone initially induced tumor regression [37]. In contrast, CBD-IL-12 in combination with CPI elicited a longer-lasting anti-tumor response that resulted in 7 CR out of 12 treated mice (58%), which indicated an important role that CBD-IL-12 may play in potentiating CPI immunotherapy for immunologically cold tumors [37]. Despite the impressive outcome of personalized autologous cancer vaccines, their manufacturing timeline and mass production limit their clinical application. Mesenchymal stromal cells (MSCs) engineered to express the immunoproteasome complex (MSC-IPr) strongly enhanced the de novo production of IL-12 and showed potent re-activation of T cell immunity against B16 melanoma, suggesting its potential as a whole-cell-based cancer vaccine, while this vaccination combined with anti-PD-1 and anti-4-1BB reached a 100% mouse survival, which revealed that 8 of 10 animals achieved complete tumor elimination [38]. A DNA vaccine coding for gp100 together with IL-12 in combination with CTLA-4/PD-1 blockade treatment significantly inhibited tumor growth in the B16F10-OVA mice model [39]. Dendritic cells stimulated with a plasmid containing the IL-12 gene significantly enhanced the anti-tumor effect of PD-1 monoclonal antibody therapy in melanoma-bearing mice [40]. Although IL-12 induces anti-tumor immunity in mouse models, the outcomes from early clinical trials using systemic recombinant IL-12 were unsatisfying with limited efficacy [41]. In Hewitt’s study, a novel intratumoral IL-12 mRNA therapy was designed to induce anti-tumor immunity with the combination of anti-PD-L1 in a model resistant to PD-L1 blockade monotherapy [41]. A phase II clinical trial (NCT 02493361) that included 23 patients with melanoma was conducted during 2018–2020, in which patients with advanced melanoma were treated with the combination of intratumoral plasmid IL-12 electroporation and pembrolizumab (PD-1 inhibitor) [36]. This study showed a better outcome of a 41% objective response rate (ORR, RECIST 1.1) and 36% complete responses (CR) as well as a well-tolerant safety [36]. IL-12 therapies for melanoma in combination with ICIs are shown in Table 2.

### 3.3. IL-12 Therapy in Combination with Other Cytokines

Cytokines are important biomolecules in the crosstalk among immune cells [42]. Even though the antitumor activities of cytokines have been revealed in many studies, it still remains a problem that the administration of a cytokine working alone can hardly induce complete tumor regression. In fact, combined usage of cytokines often generates synergistic effects [43]. IL-12 therapies of melanoma in combination with other cytokines are summarized in Table 3. As a growth factor, IL-2 is widely used clinically for its functions of activating and promoting the expansion of CD8^+^ T cells and NK cells [44]. The FDA has approved the clinical usage of high-dose of IL-2 for the treatment of patients with metastatic melanoma [45], but at the same time, the induced severe systemic toxicity, including hypotension, capillary leak syndrome, oliguric renal failure and hypoxia was reported [46]; therefore, it is necessary to transit to other interleukins for obtaining the benefits but with fewer adverse events. IL-12 can be used in combination with IL-2 to successfully control metastatic melanoma progression. In the B16.F10 murine melanoma model, combined treatment of tumors using IL-2 encoding plasmid (pIL-2) and IL-12 encoding plasmid (pIL-12) induced significant tumor growth delay and 71% complete tumor regression [47]. Mirjacic Martinovic et al. isolated peripheral blood lymphocytes from metastatic melanoma patients whose lactate dehydrogenase serum levels were normal and stimulated the cells in vitro with IL-12 and IL-2 alone as well as their combination. It was found that the combined treatment of IL-2 and IL-12 showed better outcomes than IL-12 alone in inducing the NK cell cytotoxicity [46]. One year later, a similar conclusion was made regardless of the lactate dehydrogenase level of the patients [25]. IL-12 has great potential to be used as a promising cytokine which could be combined with IL-2 in a cancer immunotherapy strategy [25].

Other than IL-2, IL-18 was also found to be effective to induce the NK cell activity and the expression of CD107a degranulation marker both in metastatic melanoma and control, when used in combination with IL-12 in peripheral blood mononuclear cells from both groups [48]. The combination of IL-12/IL-18 administration was also tested in grey horses bearing melanoma; meanwhile, increased levels of TNF-α and IFN-γ secreted from peripheral mononuclear blood cells extracorporeally and decreased levels of IL-10 secretion were observed [49,50]. Recently, novel lipid nanoparticles (LNPs) encapsulated with mRNAs encoding cytokines including IL-12, IL-27 and GM-CSF were designed to elucidate a synergistic effect in suppressing tumor growth but not causing adverse reactions compared to GM-CSF or IL-27 mRNAs in monotherapy strategies [51]. Similarly, an engineered oncolytic HSV-1 was designed to express IL-12 and GM-CSF and such a combination effectively diminished tumor growth as well as augmented the survival rate compared to treatment with oncolytic HSV-1 expressing GM-CSF (Δ6/GM) or oncolytic HSV-1 expressing IL-12 (Δ6/IL12) alone [52]. Genetic modification of immune cells (NK cells, CD8^+^ T cells) serves as a promising approach for the adoptive cell therapy of cancer patients. Yang and colleagues found that treating genetically modified CD8^+^ T cells using anti-tumor T cell receptors (TCR) with different combinations of interleukins (IL-12 plus IL-7 or IL-21) is capable of re-programming the late effector cells from TILs isolated from patients with melanoma [53]. Such combinations are named “cytokine cocktails” (the order of administration of cytokines are: 1) Initial stimulation in culture with IL-2; 2) Medium is changed to IL-12 plus IL-7 or IL-21 for 3 days; 3) Culture the cells without IL-12 for another 3~5 days). In their study, “cytokine cocktails” successfully regulate the differentiation of in vitro cultured human anti-tumor CD8^+^ T cells [53]. The combination of IL-12 with chemokine CXCL9 remarkably increased the frequency of proliferating CD8^+^ T cells in the peripheral blood, only the patients responding to the combination therapy showed significantly higher intratumoral expression of CXCR3. Meanwhile, the sensitivity of melanoma to anti-PD-1 therapy was also increased by the combination [54]. Co-expression of IL-12 and TNF-α was designed as an antitumor in situ vaccination and showed extensive infiltration of immune cells in the murine melanoma model [55]. IL-12 therapies for melanoma in combination with other cytokines are shown in Table 3.

**Table 3 cancers-14-05592-t003:** IL-12 in combination with other cytokines for therapy of melanomas.

Year	Reagents	Treatments	Objects	Results	Reference
2015	IL-18	IL-12 with IL-18	PBMC from patients with melanoma	NK cell activity↑	[48]
2015	IL-18	Linear DNA encoding IL-12/ IL-18 injection	Grey Horse	Total leukocyte and neutrophil counts↑, lymphocyte numbers↓	[50]
2016	IL-2	IL-12 with IL-2	PBMC from patients with melanoma	IL-2Rα and IL-12R expression↑, NK cell activity↑	[46]
2017	IL-2	IL-12 with IL-2	PBMC from patients with melanoma	CD4^+^CD25^bright^CD27^+^ Treg cells↓	[25]
2018	TNF-α	IL-12 and TNF-α co-expression in situ vaccination	Murine B16F10 melanoma model	Tumor growth↓, local effectiveness from 80 to 100%	[55]
2021	GM-CSF	IL-12 and GM-CSF co-expression	Murine B16F10 melanoma model	Tumor growth↓, CD4^+^ and CD8^+^ T cell recruitment↑	[52]
2022	CXCL9	IL-12 plasmid with CXCL9 plasmid	Murine B16F10 melanoma model	Tumor growth↓, DC licensing↑, CD8^+^ T cell↑	[54]
2022	IL-27 GM-CSF	LNP encapsulated with mRNAs encoding cytokines including IL-12, IL-27 and GM-CSF	Murine B16F10 melanoma model	Tumor growth↓, NK and CD8^+^ T cells↑	[51]

GET: gene electrotransfer; M1: M1 macrophages; LNP: lipid nanoparticles; ↓: decreased; ↑: increased.

### 3.4. IL-12 in Combination with Other Therapeutic Reagents

Despite the fact that IL-12 has been known as a positive regulator during the anti-tumor immune response, it is still hard to reach full remission during the process of clinical treatment for melanoma. Bleomycin, oxaliplatin and cisplatin are widely used chemotherapeutic drugs for melanoma [56,57,58]. When these chemotherapeutic agents are being used in combination with a plasmid encoding IL-12 which is peritumoral gene electrotransferred (p. t. IL-12 GET), it showed different outcomes on animals with different tumor immune status. Ursic and colleagues found that IL-12 potentiated the antitumor effect of electrochemotherapy (ECT) with biologically equivalent low doses of oxaliplatin, bleomycin, or cisplatin in B16F10 melanoma with weak immunogenicity, and after the treatment with ECT using cisplatin, the most pronounced potentiation was observed [59]. Meanwhile, better responsiveness to ECT was observed in tumors (not melanoma) with strong immunogenicity, which indicated that the tumor’s immune status effectively influences the outcome of combination therapy [59]. Similarly, using gene electrotransfer of plasmid encoding canine IL-12 in combination with bleomycin and cytoreductive surgery altogether showed a significant decline in the percentage of Treg in the peripheral blood with an OR of 67% (6/9) [60].

The vascular endothelial growth factor (VEGF) has been recognized as a critical mediator of immune suppression, implying that VEGF blockade is effective for the treatment of cancers [61]. Oncolytic adenovirus (Ad) encoding both VEGF-specific and IL-12 short hairpin ribonucleic acid (shVEGF; RdB/IL12/shVEGF) was introduced in the study [62]. Intratumoral injection of RdB/IL12/shVEGF generates massive infiltration of differentiated CD8^+^ and CD4^+^ T cells, DCs and NK cells to tissues surrounding the necrotic region of the tumor, inducing a strong antitumor effect in an immune-competent B16-F10 melanoma model [62]. Similarly, blockage of VEGF receptor-2 processed in combination with IL-12 resulted in the regression of melanoma in mice models [63]. A combination of IL-12 with a VEGF signaling inhibitor could be an excellent therapeutic strategy to achieve a better therapeutic effect due to the restoration of antitumor immune function in melanoma. Co-transfection of IL-12 and salmosin genes using anti-EGFR immunolipoplexes significantly reduced tumor growth as well as pulmonary metastasis due to its high binding affinity to the EGFR-positive cancer cells [64].

When being co-expressed with the membrane-anchored anti-CD3, IL-12 significantly increases the frequency of CD3^+^ TIL, and a broad range of T cell subsets, including activated tumor-specific T cells as well as less suppressive Treg and functional bystander T cells were observed as cellular compositions within this infiltration [65]. A clinical trial was also included in this research which was aforementioned in the “Clinical IL-12 administration in patients with melanoma” section [65]. Applying an adenovirus vector encoding IL-12 in combination with the administration of a hydrophobic TGF-β inhibitor intratumorally with a multifunctional polymer significantly increased the frequency of CD4^+^ and CD8^+^ T cells, NK cells as well as IFN-γ secretion in the tumor microenvironment, which delayed growth of B16 melanoma xenografts in mice and increased animal survival [66]. Glucopyranosyl lipid A (GLA) is known as a TLR4 agonist that induced systemic memory T cell responses when combined use with an IL-12-expressing vector, providing significant survival benefits in mouse models with melanoma [67]. IL-12 therapies for melanoma in combination with other therapeutic reagents are shown in Table 4.

There are still questions that require further elucidation regarding IL-12 combination therapies; with increasing numbers of preclinical and clinical studies with impressive outcomes, the usage of IL-12 in melanoma therapy is a promising strategy.

## 4. Strategies to Reduce the Immune-Related Side Effects of IL-12 Therapy

Although IL-12 has been considered a potent cytokine in mediating antitumor activity [68], its toxic side effects cannot be ignored and continuous efforts to improve its effectiveness and safety should be made.

### 4.1. Immune-Related Side Effects of IL-12 Therapy

Despite the strong therapeutic effect of IL-12 on different types of cancer, its toxicity prevails over its antitumor effect via dose-limiting initiating irAE [69]. A variety of studies on cancers with different types of models have shown unavoidable side effects of IL-12 administration, especially hepatotoxicity, and a high concentration of IL-12 was found in the liver after treatment of animals with IL-12 [70,71]. IL-12 irAE includes hematologic toxicity, inflammatory liver steatosis, flu-like symptoms (such as fever, and muscle pain), mucus membrane inflammation, bone marrow hyperplasia, etc. The detailed introduction of IL-12 irAE was seen in Lasek’s review [72]. irAE is not only shown in clinical trials of IL-12 administration on patients with cancers but also shown in mice and squirrel monkey models are summarized in Figure 3.

### 4.2. Strategies to Improve the Effectiveness and Safety

To improve the anti-tumor efficacy and reduce the irAE of IL-12, several groups have tested modified vectors for the expression of IL-12 or novel strategies to deliver IL-12. Kamensek et al. evaluated the IL-12 fusion gene expression plasmid without an antibiotic resistance gene in a preclinical B16F10 mouse melanoma model and showed a good expression and safety profile of the p21-hIL-12-ORT GET as well as the promising antitumor efficacy [73]. The immunotoxicity of IL-12 can be masked by fusing it with a domain of its receptor (M-L6-IL-12) via a tumor-protease-cleavable linker, thus no irAE was observed in a mouse model with melanoma. Moreover, when combined with anti-PD-1 treatment, M-L6-IL-12 resulted in significantly extended survival [74]. An improved plasmid encoding IL-12p70 incorporating a picornavirus-derived co-translational P2A site was developed in 2017 which showed higher transfection efficiency and expression of plasmid-derived IL-12p70, as well as its downstream effector IFN-γ [75]. Galvan et al. developed an engineered three piggyBac transposon vector expressing mIL-12 (p35/p40 IL-12) and modified mouse splenocytes with it, which efficiently improved the survival of the melanoma mouse [76]. Engineered T cells expressing membrane-anchored IL-12 (aIL-12) using a transmembrane (TM) anchor domain were developed and applied in a human tumor xenograft model, and significant augmentation of the tumor regression mediated by human T cells was observed [77,78]. Hewitt et al. designed an intratumoral IL-12 mRNA therapy strategy that linked the mRNA sequence of mIL-12 and MED11191 to generate a linked monomeric IL-12p70 (IL-12B-IL-12A), a single intratumoral dose of which effectively induced IFN-γ and CD8^+^ T cell-dependent tumor regression in a mouse model with melanoma as well as reducing undesirable toxicity [41]. By treating the B16 tumor-bearing mice with IL-12 cDNA via hydrodynamic gene transfer, Savid-Frontera C et al. demonstrated that a single administration of 1 μg of IL-12 cDNA is capable of inducing tolerated systemic IL-12 and a potent immune response as well as higher control of tumor growth [79]. Intratumoral gene electroporation uses electric charges to help plasmid DNA enter target cells reproductively and effectively, especially in metastatic melanomas [33]. Other than gene electroporation, lipid nanoparticles [51], dual-targeting nanoparticles [80], hypoxia-responsive nanogel [81], cationic polyphosphazene vesicles [82], collagen [37] and heparin-based complex coacervate [23] were also developed to allow more efficient delivery as IL-12 or IL-12 encoding plasmid carrier, despite the artificial chemicals, mesenchymal stromal cells [83] and DCs [84] overexpressing IL-12 exert therapeutic properties of IL-12 cytokine without toxicity. Great efforts should be made in the future to further improve the anti-tumor efficacy and diminish the side effects of IL-12 therapy.

## 5. Concluding Remarks and Future Perspectives

Cytokine-based therapy has shown great potential for durable antitumor responses in melanoma. IL-12 comprises key elements for sustaining the crosstalk between tumor and immune cells as they are essential in cytokine-related signaling networks. Although IL-12 was proven to have great potential for clinical usage in melanoma patients, its toxic side effects, including hematologic and hepatic toxicity, cannot be ignored. In the past ten years, promising technological approaches such as newly developed carriers and engineered immunocytes have improved the anti-tumor activity and safety in in vivo and ex vivo preclinical studies. However, clinical trials are needed to further evaluate their effectiveness and safety in IL-12 immunotherapy. It was found that IL-12 could inhibit the expression of PD-1, LAG3 in activated T cells and PD-L1 in tumor cells [18,30], which suggests a promising benefit in immune checkpoint therapy. Therefore, it is worth evaluating the anti-tumor efficacy of IL-12 therapy in combination with immune checkpoint therapy in more preclinical and clinical studies. Furthermore, the role of IL-12 in engineered immune cell (such as CAR-T, CAR-NK, TCR-T) therapy also deserves further evaluation. These studies will help us better understand the mechanisms and improve the strategy for immunotherapy of cancers.

## Figures and Tables

**Figure 1 cancers-14-05592-f001:**
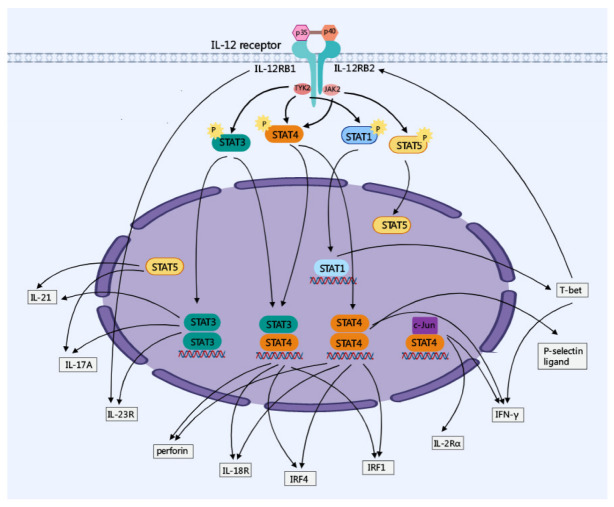
IL-12-related signaling pathway. IL-12 is unique in having the only heterodimeric cytokines, which are composed of an α-chain (p35) and a β-chain (p40). IL-12 signals via its receptor IL-12RB1 and IL-12RB2 are mediated by members of the JAK-STAT family which mediate the expression of downstream proteins, such as IL-21, IL-17A, IL-23R, IL-18R, IL-2Rα, T-bet and IFN-γ.

**Figure 2 cancers-14-05592-f002:**
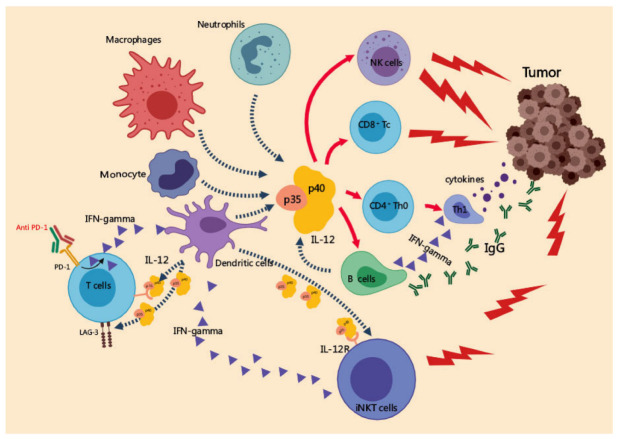
Anti-tumor mechanisms of IL-12. On the one hand, IL-12 secreted by APCs promotes the polarization of Th1 cells and the cytotoxic activity of CD8^+^ T cells, NK cells, and NKT cells. On the other hand, IL-12 inhibits the differentiation of Tregs and the expression of PD-1, LAG3 in activated T cells and the PD-L1 on tumor cells, thereby preventing the cytotoxic T cells from negative feedback inhibition.

**Figure 3 cancers-14-05592-f003:**
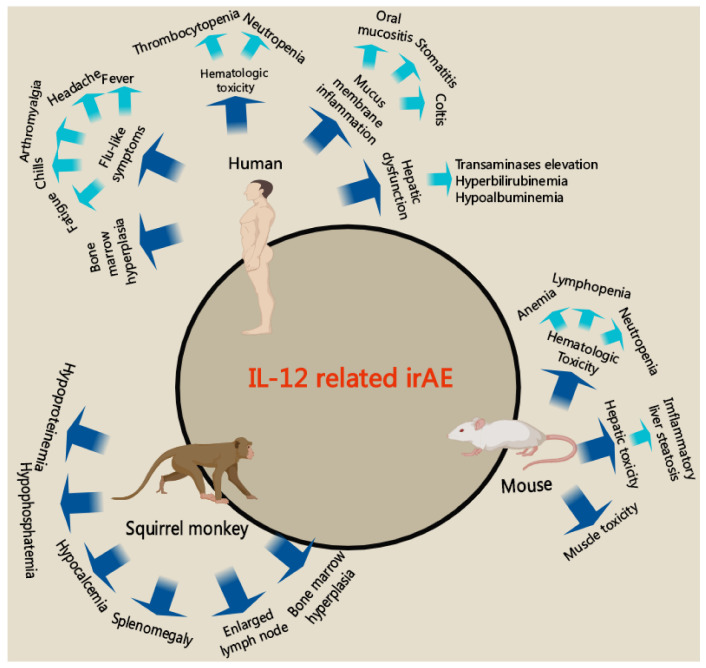
IL-12 related irAE. In clinical trials of patients with cancer, hematologic toxicity, inflammatory liver steatosis, flu-like symptoms, mucus membrane inflammation, bone marrow hyperplasia are found the major side effects. In preclinical experiments with mouse tumor models, hematologic toxicity, inflammatory liver steatosis and muscle toxicity are seen. However, hypoproteinemia, hypophosphatemia, hypocalcemia, splenomegaly, enlarged lymph node and bone marrow hyperplasia are the most common irAE seen in squirrel monkeys.

**Table 1 cancers-14-05592-t001:** Clinical trials in melanoma patients with IL-12 therapy since 2012.

Year	NCT Number/ Study Phase	Treatment	Patients Number	Outcomes	Reference
2013	NCT00683670 Phase I	Vaccination with CD40L/IFN-γ–matured, IL-12p70–producing DCs	7	One CR, two PR, one SD in melanoma patients	[31]
2015	NCI 1-C-011 Phase I	Autologous TIL transduced with a gene encoding a single chain IL-12 driven by a nuclear factor of activated T cells promoter (NFAT.IL12)	33	One CR, nine PR in melanoma patients	[32]
2020	NCT 01502293 Phase II	Intratumorally with plasmid encoding IL-12 (tavokinogene telseplasmid; tavo), 0.5 mg/mL followed by electroporation (six pulses, 1500 V/cm)	30	Five CR, seven PD in melanoma patients	[34]
2020	NCT02493361 Phase II	Tavo was administered intratumorally days 1, 5, and 8 every 6 weeks while pembrolizumab (200 mg, i.v.) was administered every 3 weeks.	23	Nine CR, two PR, three SD, nine PD in melanoma patients	[36]

**Table 2 cancers-14-05592-t002:** IL-12 in combination with ICI for therapy of melanomas.

Year	Reagents	Treatments	Objects	Results	Reference
2019	Anti- PD-1 anti-CTLA-4	IL-12 plasmid intradermal injection with anti- PD-1 or anti-CTLA-4	Murine B16F10 melanoma model	Tumor growth↓	[39]
2020	Pembrolizumab	Tavo encoding IL-12 with pembrolizumab	Patients with melanoma	Immune infiltration↑, ORR 48% (11/23)	[36]
2020	Anti-PD-L1	Modified CD8^+^ T cells with mIL-12 mRNA with anti-PD-L1	Murine B16F10 melanoma model	Anti-tumor immunity↑, tumor growth↓, un-injected distal lesions↓	[41]
2020	Anti-PD-1 anti-CTLA-4	A collagen-binding domain fused to IL-12 with anti- PD-1 or anti-CTLA-4	Murine B16F10 melanoma model	Tumor growth↓, antigen-specific immunological memory↑	[37]

↓: decreased; ↑: increased.

**Table 4 cancers-14-05592-t004:** IL-12 in combination with other therapeutic reagents.

Year	Reagents	Treatments	Objects	Results	Reference
1998	Cisplatin	IL-12 plasmid with cisplatin	Murine B16F10 melanoma	No significant improvement	[58]
2012	VEGFR-2	T cells cotransduced with an anti-VEGFR- 2 CAR and a constitutively expressed single-chain murine IL-12 or an inducible IL-12 gene after host lymphodepletion	Murine B16F10 melanoma model	Tumor growth↓ CD11b^+^Gr1^+^ cells↓	[63]
2016	shVEGF	Oncolytic adenovirus co-expressing IL-12 and VEGF-shRNA	Murine B16F10 melanoma model	Tumor growth↓ CD4^+^, CD8^+^ T cells↑, NK cells and DCs↑	[62]
2017	TGF-β inhibitor	IL-12 plasmid and SB-505124	Murine B16F10 melanoma model	Tumor growth↓ CD4^+^, CD8^+^ T and NK cells↑	[66]
2021	GLA formulated in a stable emulsion	IL-12 plasmid and GLA-SE	Murine B16F10 melanoma model	Tumor growth↓, CD8^+^ T cells↑	[67]
2022	Membrane- anchored anti-CD3	Electroporation of IL-12 and membrane-anchored anti-CD3 plasmids	Patients with unresectable, stage III/IV melanomas	Restored the function of TIL isolated from a patient with melanoma actively progressing on PD-1 blockage	[65]

GLA: glucopyranosyl lipid A; ↓: decreased; ↑: increased.

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
