# Peer review of "Antitumor Activities of Interleukin-12 in Melanoma"

_cancers, 2022, doi:10.3390/cancers14225592_

Round 1

Reviewer 1 Report

This is a review about antitumor activities of IL-12 in melanoma, with focus on melanoma treatment. The structure of the review is excellent. There are some mistakes in grammar, but those can be easily corrected. 

For example line 88 Though the precise mechanisms underlying IL-12 antitumor effects need to be further clarify (should be 'clarified') . 

Sometimes the authors are not precise. 

For example: line 69 - more production of Th1-associated classes of immunoglobulin [15]. (do you mean class switching enhanced by Th1-cells?)

Although I believe the authors to be thorough, I personally always suggest a 'systematic review approach' to make sure no literature is overlooked. In this review for example it would help to show in the supplement the search strategy and how it was made sure all relevant literature, relating treatment, was indeed assessed. So it does not have to relate all IL-12 literature, but it is nice to make sure that all trials (including mouse models) were indeed described. 

I am very fond of the images! 

Author Response

Reviewer#1This is a review about antitumor activities of IL-12 in melanoma, with focus on melanoma treatment. The structure of the review is excellent. There are some mistakes in grammar, but those can be easily corrected. 

1. For example, line 88 Though the precise mechanisms underlying IL-12 antitumor effects need to be further clarify (should be 'clarified').

Reply: Thank you for your positive comments on our manuscript. “Clarify” was replaced with “clarified” (line #97), other corrections were also made with track changes (Line #87, #109, #141, #177, #348)

2. Sometimes the authors are not precise. For example: line 69 - more production of Th1-associated classes of immunoglobulin [15]. (do you mean class switching enhanced by Th1-cells?)

Reply:To avoid unnecessary misunderstanding, we changed the sentence into “Directly or with the help of IFN-γ, IL-12 activates the immune function of B cells with more production of IgG2a [15]” (Line #76-77)  

3. Although I believe the authors to be thorough, I personally always suggest a 'systematic review approach' to make sure no literature is overlooked. In this review for example it would help to show in the supplement the search strategy and how it was made sure all relevant literature, relating treatment, was indeed assessed. So it does not have to relate all IL-12 literature, but it is nice to make sure that all trials (including mouse models) were indeed described. 

Reply: Our search strategy: Search: (((IL-12) AND (melanoma)) AND (therapy)) AND (("2012"[Date - Publication] : "2022"[Date - Publication])) , 248 publications were shown. We excluded the articles that is not relevant to IL-12 treatment. We make sure that all related clinical trials and in vivo studies were included in our review.

Please see the attachment (cover letter to editor and reviewers)

Reviewer 2 Report

The review article “Antitumor Activities of Interleukin-12 in Melanoma” (by Gao W et al.) describes the attempts to treat advanced melanomas with interleukin-12 (IL-12), alone or with combination with other cytokines or antitumor compounds. The authors describe also the problem with the IL-12 side effect. The authors worked on IL-12, but there is only limited number of their papers which connect IL-12 and melanoma. The topic is rather narrow but the text gives useful information to readers. There are no specific concerns, some small corrections in English or formulations would improve the text.

Author Response

The review article “Antitumor Activities of Interleukin-12 in Melanoma” (by Gao W et al.) describes the attempts to treat advanced melanomas with interleukin-12 (IL-12), alone or with combination with other cytokines or antitumor compounds. The authors describe also the problem with the IL-12 side effect. The authors worked on IL-12, but there is only limited number of their papers which connect IL-12 and melanoma. The topic is rather narrow but the text gives useful information to readers. There are no specific concerns, some small corrections in English or formulations would improve the text.

Reply:We really appreciate your careful review and positive comments. We have polished the language and made several corrections (Line #87, #109, #141, #177, #348).

Please see the attachment (cover letter to editor and reviewers).
